# HtmlRAG: HTML is Better Than Plain Text for Modeling Retrieved Knowledge in RAG Systems

## Abstract

Retrieval-Augmented Generation (RAG) has been shown to improve knowledge capabilities and alleviate the hallucination problem of LLMs. The Web is a major source of external knowledge used in RAG systems, and many commercial systems such as ChatGPT and Perplexity have used Web search engines as their major retrieval systems. Typically, such RAG systems retrieve search results, download HTML sources of the results, and then extract plain texts from the HTML sources. Plain text documents or chunks are fed into the LLMs to augment the generation. However, much of the structural and semantic information inherent in HTML, such as headings and table structures, is lost during this plain-text-based RAG process. To alleviate this problem, we propose HtmlRAG, which uses HTML instead of plain text as the format of retrieved knowledge in RAG. We believe HTML is better than plain text in modeling knowledge in external documents, and most LLMs possess robust capacities to understand HTML. However, utilizing HTML presents new challenges. HTML contains additional content such as tags, JavaScript, and CSS specifications, which bring extra input tokens and noise to the RAG system. To address this issue, we propose HTML cleaning, compression, and pruning strategies, to shorten the HTML while minimizing the loss of information. Specifically, we design a two-step block-tree-based pruning method that prunes useless HTML blocks and keeps only the relevant part of the HTML. Experiments on six QA datasets confirm the superiority of using HTML in RAG systems [1].

## CCS Concepts

• **Information systems** → **Web search engines**.

## Keywords

HTML, Retrieval-Augmented Generation, Large Language Model

**ACM Reference Format:**
Anonymous Author(s). 2018. HtmlRAG: HTML is Better Than Plain Text for Modeling Retrieved Knowledge in RAG Systems. In *Proceedings of TheWebConf 2025 (Conference acronym 'XX).* ACM, New York, NY, USA, 13 pages. https://doi.org/XXXXXXX.XXXXXXX

---

[1]Code and datasets are available at https://anonymous.4open.science/r/HtmlRAG-0F8C

---

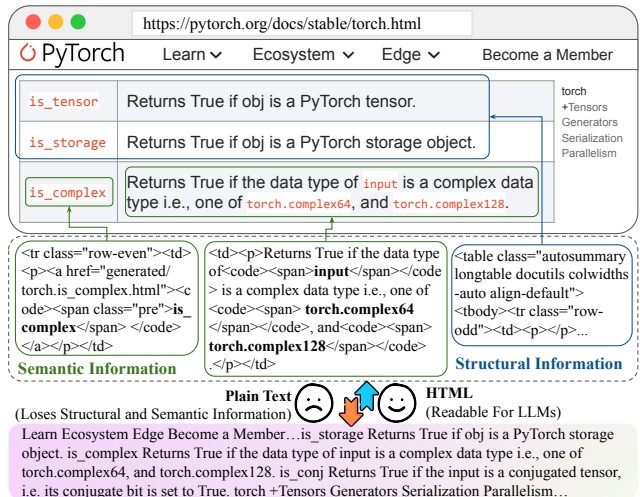

**Figure 1: Information loss in HTML to plain text conversion.**

## 1 Introduction

Large Language Models (LLMs) have been proven to have powerful capabilities in various natural language processing tasks [42, 44, 46]. However, at the same time, LLMs show deficiencies such as forgetting long-tailed knowledge [28], offering outdated knowledge [3], and hallucination [38, 39, 74]. Retrieval-augmented generation (RAG) utilizes a retrieval system to fetch external knowledge and augment the LLM. It has proved effective in mitigating hallucinations of LLMs [41, 76]. Many RAG systems, such as Perlexity [47] and SearchGPT [43], have been developed, and they commonly use Web search engines as the underlying retrieval systems.

Traditional RAG pipelines typically use plain text as the format for retrieved knowledge [21, 63]. HTML documents from the Web are often converted into plain text and concatenated with the user's query before being fed into the LLM. We found that converting HTML to plain text leads to the loss of structural and semantic information. Figure 1 illustrates that a web page containing tabular form becomes disordered when converted to plain text. Even worse, original HTML tags, such as "" and "<a>", denoting important information, are discarded during conversion. Thus, in this paper, we tend to investigate an intuitive idea: *Can we take HTML as the format of external knowledge in RAG systems to preserve the information in HTML documents to a larger extent?*

Taking HTML as the format of external knowledge offers several advantages beyond preserving the information inherent in HTML documents. During pre-training, LLMs have encountered HTML documents [6, 15, 17], which means that they inherently possess the ability to understand HTML without requiring further fine-tuning [26, 73]. Recently, both proprietary and open source LLMs have begun to support increasingly longer input windows, making

it feasible to input more extensive HTML documents [11, 69, 72]. Furthermore, documents in Latex, PDF, and Word formats can be converted to HTML with minimal loss, expanding the potential application of HTML as the format of external knowledge [7, 61, 64].

However, employing HTML as the knowledge format for LLMs also presents the challenge of handling longer input sequences and noisy contexts. Our preliminary experiments show that a real HTML document from the Web contains over 80K tokens on average, among which over 90% of the tokens are CSS styles, JavaScript, Comments, or other meaningless tokens. Compared to the common maximum context window of current LLMs, which ranges from 32K to 128K, an individual document length of 80K is unacceptable. The noisy tokens. The aforementioned meaningless tokens in HTML documents can also affect the generation quality of LLMs. To solve this problem, in this paper, we devise a **HTML Cleaning** module to remove semantically irrelevant content in HTML documents, while keeping the main content intact. We also adjust the HTML tree structure without losing semantic information, for example, merging multiple layers of single nested HTML tags and removing empty tags. These processes reduce the length of the HTML to 6% of its original size.

Even after cleaning, HTML documents remain relatively long (over 4K each) to LLMs. To shorten the input context and remove the noise contained in the original retrieved documents, existing RAG systems have utilized different types of post-retrieval result refiners [19, 22, 66, 75]. These refiners extract the relevant text chunks or key sentences from the documents, regarding the user's query and LLMs' preference, and discard other content. These plain-text-based refiners cannot be directly applied to HTML because simply chunking HTML without considering its structure may generate unreasonable chunks. Hence, we further design an **HTML Pruning** module, which functions upon the intrinsic tree structure of HTML. The pruning process is comprised of the following steps:

(1) **Building a Block Tree**. Each HTML document can be parsed into a DOM tree [58]. We do not simply prune HTML on the DOM tree because it is too finely-grained [16, 62], which brings much computational cost. Instead, we propose to build a corresponding block tree, in which the original DOM tree nodes are merged into hierarchical blocks. The granularity of the block tree can be adjusted by the degree of merging.

(2) **Pruning Blocks based on Text Embedding**. We then prune the block tree using an on-the-shelf embedding model, because it is a simple but effective way to calculate the block's relevance scores with the user's query based on their embedding similarity. We apply a greedy pruning algorithm that removes blocks with lower similarity scores, and gets a pruned block tree. However, we observe that the embedding model may fail to work well with the fine-grained blocks because embeddings learned for these small blocks are usually vague and inaccurate, so this pruning step is limited to coarse-grained block trees.

(3) **Generative Fine-grained Block Pruning**. To prune the block tree further, we expand the leaf nodes of the pruned block tree and build a finer-grained block tree. Since the generative model has a longer context window, it can model the block tree globally and is not limited to modeling one block at a time. Thus we further develop a generative model to prune HTML over the fine-grained blocks. The generative model is supposed to calculate the score for each block, which is given by the generation probability of a unique sequence indicating the block. The sequence is given by the path of HTML tags, starting from the root tag and walking down to the block's tag and text (e.g., "<html><body><div><p>block content..."). Finally, according to the block scores, we apply a similar greedy pruning algorithm to get the final pruned HTML.

We conduct extensive experiments on six datasets including ambiguous QA, natural QA, multi-hop QA, and long-form QA. Experimental results confirm the superiority of HTML as the format of external knowledge over plain text.

Our contributions are threefold: (1) We propose to take HTML as the format of knowledge in RAG systems, which retains information of the original HTML; (2) We propose a simple but effective HTML cleaning algorithm; (3) We propose a two-stage HTML pruning algorithm. This can be applied to most RAG systems and strikes a balance between efficiency and effectiveness.

## 2 Related Works
### 2.1 Retrieval-Augmented Generation (RAG)
RAG systems augment LLM with external knowledge. A typical RAG pipeline includes components such as a query rewriter [55], a retriever [32, 53], a reranker [53, 63], a refiner [19, 22, 66], and a reader [5, 77]. This typical pipeline is widely used by mainstream RAG frameworks, such as LangChain [8] and LlamaIndex [35]. Many works aim to optimize components in the pipeline, and previous works also manage to enhance the performance of RAG in other ways. Some methods devise new RAG frameworks, like retrieving external knowledge actively when internal knowledge is missing [5, 20, 55], or letting the LLM plan the retrieval process in a straight line or a tree structure [27, 52]. However, most existing RAG systems take plain text as the format of external knowledge. Instead, we propose to take HTML as the format of external knowledge, and we believe using HTML can keep richer semantics in retrieved results.

### 2.2 Post-Retrieval Process of RAG
RAG systems usually apply post-retrieval processes (i.e., result refiners) to extract only the useful content to shorten the input context sent to LLMs. The chunking-based refiner is a widely used solution, which first chunks the text according to certain rules, and then uses a reranking model to select top chunks with high relevance [25, 40]. Another solution is abstractive refiner, which utilizes a text-to-text language model to generate abstracts of results [14, 19, 66]. Some works use off-the-shelf abstractive models [70, 71] or fine-tuned abstractive models [19] to summarize retrieved results in a segmented and hierarchical manner. Others leverage the logits of language models to determine the importance of words within documents [33, 37].

The aforementioned post-retrieval result refiners are all based on plain text. The existing chunking-based methods cannot be directly applied to HTML because simply chunking HTML without considering its structure may generate unreasonable chunks. Furthermore, the abstractive refiners may have problems such as difficulty in dealing with excessively long HTML, high computational cost, or limited understanding of HTML. To alleviate these problems, in this paper, we propose to prune HTML based on its DOM structure.

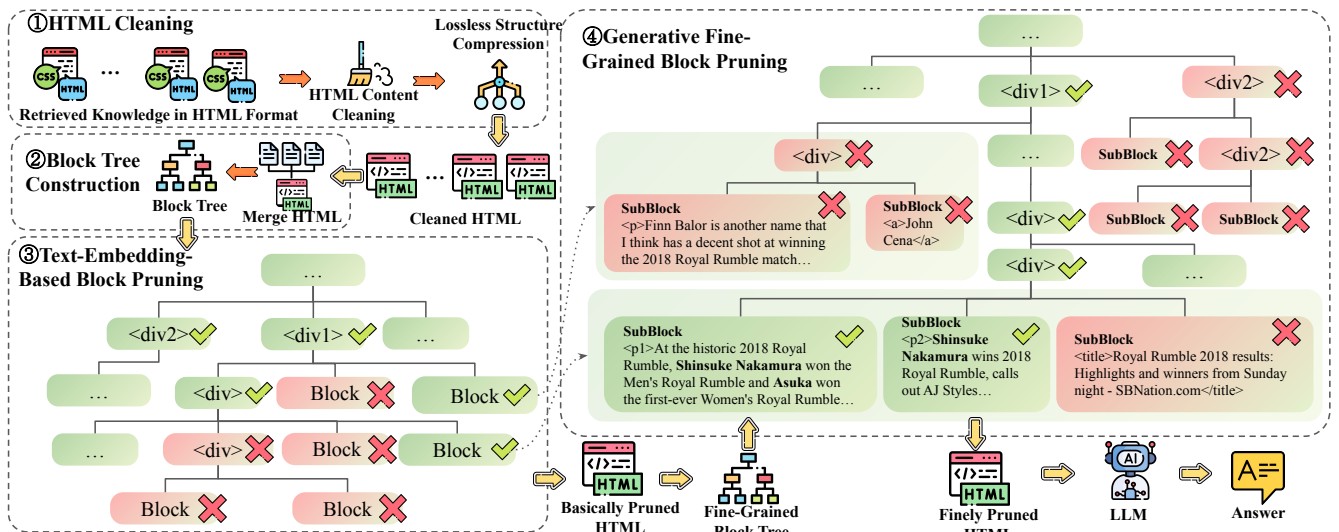

**Figure 2: HTML for RAG pipeline overview**

## 2.3 Structured Data Understanding

Previous works have demonstrated that structured data such as HTML [9, 68] and Excel tables [31, 57, 59] contain richer information compared to plain text. These works design specialized tasks [30, 59] over structured data or fine-tune language models to understand structured data [4, 62]. Our research is not limited to understanding a certain format of data but recommends using a richer data format in the general RAG systems. To the best of our knowledge, we are the first to propose using HTML as the input for RAG systems.

## 3 Methodology

In this paper, we propose HtmlRAG, which uses HTML instead of plain text as the format of retrieved knowledge in RAG systems, aiming to keep richer semantic and structured information that is missing in plain text. We emphasize that HTML is a popular data format for documents in a knowledge base and other document formats can be easily converted into HTML.

Taking HTML as the format of external knowledge presents a new challenge of excessively long context. Hence, in HtmlRAG, we propose to prune the original HTML documents into shorter ones progressively. We first apply an HTML cleaning module (§3.2) to remove useless elements and tags. We then propose a two-step structure-aware pruning method to further refine the resulting HTML (§3.4). More specifically, we delete less important HTML blocks with low embedding similarities with the input query (§3.4.1), and then conduct a finer block pruning with a generative model (§3.4.2). The overview of our method is shown in Figure 2.

## 3.1 Problem Definition

In the RAG pipeline, a retriever retrieves a collection of HTML documents $D$ from the Web, with a total length of $L$. Meanwhile, we have an LLM $M$ as the reader, which generates an answer $a$. The LLM has a maximum length of context window $l$, considering both efficiency and quality. Our HTML compression algorithms map $D$ to a shorter HTML document $d$. Its length can fit into the LLM's context window, namely the length of $d$ must be less than or equal to $l$. Our goal is to optimize the compression algorithm to find the best mapping from $D$ to $d$ so that the answer $a$ output by the LLM has the highest quality.

## 3.2 HTML Cleaning

Since the original HTML documents are excessively long (over 80K each), and it's needless to involve semantic features, model-based methods are inappropriate at this step. Thus, we first design a rule-based HTML cleaning, which pre-processes the HTML without considering the user's query. This cleaning process removes irrelevant content and compresses redundant structures, retaining all semantic information in the original HTML. The compressed HTML of HTML cleaning is suitable for RAG systems equipped with long-context LLMs and are not willing to lose any information before generation. The cleaned HTML also serves as the basis for the following HTML pruning.

*3.2.1 HTML Content Cleaning.* The HTML documents retrieved from the Web contain a large amount of extra content that is invisible to human users, such as HTML tags, CSS, JavaScript, etc. Most of the HTML tags provide rich structural information that helps the LLM understand the HTML, while CSS and JavaScript content provide limited assistance. So the specific cleaning steps, which are almost lossless, are as follows: (1) We remove CSS styles, Comments, and JavaScript; (2) We clear lengthy HTML tag attributes.

*3.2.2 Lossless Structural Compression.* We find that in most HTML documents, their original HTML structure contains redundancies. We can conduct the following compression to the HTML structure without losing semantic information: (1) We merge multiple layers of single-nested tags. For example, we simplify "<div><div><p>some text</p></div></div>" to "<p>some text</p>"; (2) We removed empty tags, such as "<p></p>".

## 3.3 Granularity-Adjustable Block Tree Construction

To prune all retrieved HTML documents as a whole, we first concatenate all retrieved HTML documents together, and use Beautiful Soup [50] to parse the concatenated HTML document to a single DOM tree. Pruning HTML using the DOM tree is the most natural way, but the DOM tree is so finely-grained that numerous nodes and the deep tree structure bring huge computational costs.

Considering the above problem, we propose an optimized tree structure that models HTML, which is not so fine-grained. Ideally, the granularity of the tree structure can be adjusted for different pruning requirements. We term it as a "block tree", and we set the maximum number of words per block, $maxWords$ to control the granularity of the block tree. In terms of block tree construction, we start from a DOM tree, and we merge fragmented child nodes into their parent and treat them as a block. We can recursively merge blocks or child nodes into their parent to form a bigger block under the condition that the number of words in a block does not exceed $maxWords$. After merging, original leaf nodes that are unable to be merged are also regarded as blocks. Algorithm details are demonstrated in Appendix B.

## 3.4 Block-Tree-Based HTML Pruning

The block-tree-based HTML pruning consists of two steps, both of which are conducted on the block tree structure. The first pruning step uses an embedding model to prune the result output by the HTML cleaning module, while the second step uses a generative model to prune the result output by the first pruning step.

*3.4.1 Pruning Blocks based on Text Embedding.* The refining process is expected to shorten the retrieval results while preserving key information as much as possible. A straightforward idea is to extract plain text in the block and calculate a similarity score with the user's query using text embeddings. Then we use a greedy algorithm to prune the block tree by deleting low-similarity blocks and retraining higher ones. In practice, we keep deleting the block with the lowest relevance until the total length of the HTML documents satisfies the context window we set. After block deleting, redundant HTML structures will re-appear, so we re-adjust the HTML structure, meaning multiple layers of single-nested tags are merged and empty tags are removed. The detailed pruning algorithm is demonstrated in Appendix B.

The embedding-based HTML pruning algorithm is lightweight but effective. It adapts to the HTML format better compared to plain-text-based refiners. However, it still has limitations, mainly reflected in the following aspects: (1) The embedding model's context window is limited to the scope of text within the block each time. It does not directly compare candidate blocks in a single inference. Thus the embedding model lacks a global view of the document information; (2) The embedding model cannot handle block trees with finer granularity, because the text within most blocks is not long enough for the embedding model to obtain semantic features.

*3.4.2 Generative Fine-Grained Block Pruning.* To further prune blocks with a finer granularity, we expand the leaf nodes of the pruned block tree and get a finer-grained block tree. Given the limitations of the embedding-model-based block pruning, we propose

to use a generative model because it has a long context to cover the whole block tree and is not limited to modeling one block at a time. Yet processing the cleaned HTML directly with a generative model is inappropriate because the cleaned HTML is long (60K on average), which brings much computational cost. Similarly, the generative model is supposed to calculate scores for blocks. Inspired by CFIC [48], which takes the text chunk's sequence generation probability as the score for that chunk, we propose to use a sequence of tags to identify a block. Specifically, the sequence consists of tags starting from the root tag and walking down to the block's tag, and we term this sequence as "block path". In the inference phase, the generative model follows the structure of the block tree and calculates the scores of blocks in the block tree. The scores of blocks are derived from the token logits, as displayed in Figure 3. At last, we use the same block pruning operation as we mention in §3.4.1 to obtain the refined HTML document.

The details of the generative fine-grained block pruning module are introduced in the remaining section.

**(1) Training a Path-aware Generative Model**. Long-context LLMs are capable of modeling a long-context input containing HTML format and following instructions [10, 36]. Considering the computational cost, we employ an existing lightweight long-context LLM as the foundation model. The model input is the concatenation of an HTML, the query, and an instruction, as demonstrated in Figure 4. The instruction is specially designed to help the LLM understand this path generation task, but we find that the unfine-tuned LLM does not meet our requirements. We attribute this to the fact that existing LLMs have not encountered similar tasks or instructions in either pre-training data or instruction fine-tuning data, because the path generation task is proposed for the first time.

Thus we fine-tune the generative model to align with the target of generating the path for the most relevant block. So we design the output format as shown in Figure 4: the block path, followed by the block content. The block content is appended to provide an extra supervising signal that helps the generative model learn the features of the most relevant block. Additionally, to discriminate between children with the same tag name, we append a number to the end of the original tag name. For example, two children with the same "<div>" tag are renamed as "<div1>" and "<div2>".

We collect a small amount of supervised data to enhance the model's capability in block path generation. Following the typical SFT process [49], the steps for training data collecting, filtering, and constructing are as follows: First, we sample queries from the training set of several open-source QA datasets. For each query, we retrieve a couple of related HTML documents using the online search engine Bing. Then we clean the retrieved HTML, and prune the HTML with the embedding model. By adjusting the output length in HTML pruning, we get pruned HTML documents of various lengths, ranging from 2K tokens to 32K tokens. After that, we build a block tree from each HTML document pruned by the embedding model, and calculate the exact match score for the content within blocks with the gold answer. To ensure the data quality, we discard samples in which no block's content exactly matches the gold answer, meaning highly relevant HTML documents are not retrieved. More training details are discussed in Appendix A.

**(2) Efficient Tree-Based Inference with Dynamic Skipping**. During inference, the generative model is supposed to calculate

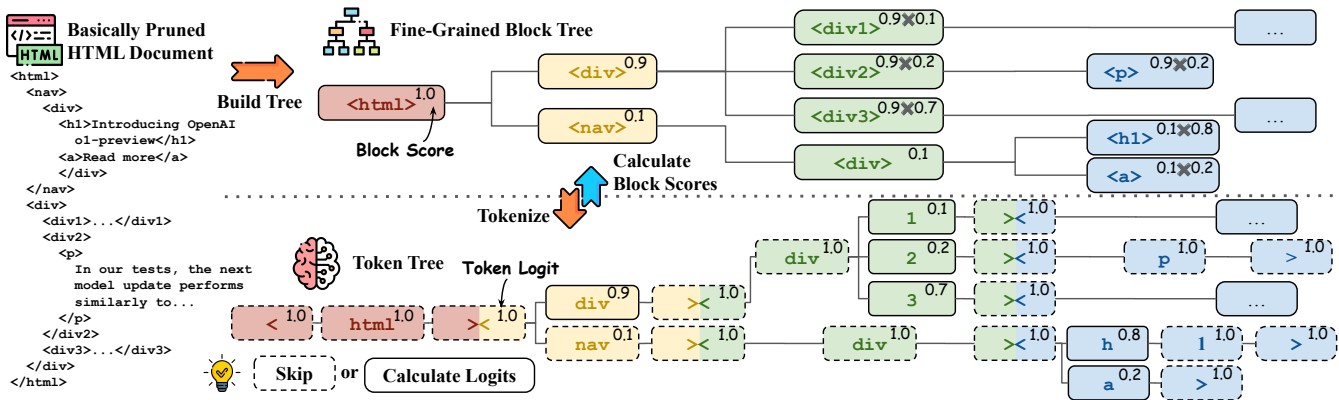

**Figure 3: Block score calculation. The block tree is transformed into the token tree with a tokenizer, and corresponding HTML tags and tokens are marked with the same colors. Token generation probabilities are in the upper right corner, and tokens in dashed boxes do not require inference. In the upper right corner of the block tree, the block probabilities are displayed, which can be derived from the corresponding token probabilities.**

```
Input:
**HTML**:  "{HTML}"
**Question**:  **{Question}**
Your task is to identify the most relevant text piece
to the given question in the HTML document. This text
piece could either be a direct paraphrase to the fact,
or a supporting evidence that can be used to infer the
fact. The overall length of the text piece should be
more than 20 words and less than 300 words. You should
provide the path to the text piece in the HTML document.
An example for the output is: <html1><body><div2><p>Some
key information...
Output:
<html1><body><div2><p>At the historic 2018 Royal Rumble,
Shinsuke Nakamura won the Men's Royal Rumble...
```

**Figure 4: The prompt for the generative model.**

block scores, and the score for block $b$ is Score($b$). Each block has a block path, and we first tokenize it to tokens $\{t_1, t_2, \cdots, t_N)\}$, suppose it has $N$ tokens in total (e.g., "<html><div>" is tokenized to {"<", "html", "><", "div", ">"}). Given the model's input sequence $input$ and $n - 1$ already generated tokens, the generative model GenModel calculates the logit of the $n$-th token $t_n$ in the output sequence as below:

$$\text{Logits}(t_n) = \text{GenModel}(t_n | \{input, t_1, \cdots, t_{n-1}\}). \quad (1)$$

We propose an efficient tree-based inference, and the tree is termed as the "token tree", which has a one-to-one correspondence with the block tree, given a specific tokenizer. We merge tokenized block paths to get the block tree, as Figure 3 shows. For example, {"<", "html", "><", "nav", ">"} and {"<", "html", "><", "div", ">"} share the same prefix, {"<", "html", "><"}, and can be merged. Ultimately, the $i$-th token in the tokenized block path will appear at the $i$-th level of the token tree. After the token tree construction, we calculate the probabilities of tokens in the token tree. The calculation

has the following conditions: (1) The probability of the root node is 1.0, which is often "<", depending on the tokenizer; (2) The probabilities of singleton child nodes, which have no siblings, are 1.0; (3) The probabilities of other nodes are calculated by the generative model $GenModel$. Suppose token $t_n$ has $K$ siblings, which are the $n$-th token in the output sequence, we get the logits of siblings $\{t_n^1, t_n^2, \cdots\}$ by Equation (1) and take the softmax of logits as probabilities. In summary, the probability of a token $t_n^k$ (the $n$-th token in the tokenized block path, and the $k$-th sibling) is given by:

$$P(t_n^k) = \begin{cases} 1.0, & \text{if } n = 1 \text{ or } K = 1; \\ \frac{\exp(\text{Logits}(t_n^k))}{\sum_{i=1}^{K} \exp(\text{Logits}(t_n^i))}, & \text{overwise.} \end{cases} \quad (2)$$

In the first two conditions, it is needless to infer with the generative model, meaning many tokens can be skipped. This brings down the inference computational cost. Apart from token skipping, the order of token logit calculation also matters a lot in computational cost. We apply a depth-first algorithm to traverse the token tree and calculate token logits so that the tokens that are calculated sequentially share the longest prefix sequence. This strategy reuses the KV cache of prefix sequences at most. Algorithm details are displayed in Appendix B.

At last, we transform the generation probabilities from the token tree back to the block tree so that we can calculate block scores. To prevent precision overflow, we take the sum of the logarithm of token probabilities as the score of the block $b$:

$$\text{Score}(b) = \sum_{i=1}^{N} \log(P(t_i)). \quad (3)$$

After we get the block scores, we reuse the greedy block pruning algorithm introduced in §3.4.1 to get the finely pruned HTML.

## 4 Experiments

We conduct experiments on six QA datasets. We simulate the real industrial working scenario for web search engines and compare our method with baselines from various paradigms.

**Table 1: Results of HtmlRAG and baselines under the short-context setting. Hit@1 is the proportion of instances where at least one short answer matches. The best and second best results are in bold and underlined. The symbol † signifies that our model achieves superior results among baselines in a statistically significant manner (t-test, $p$-value < 0.05).**

| Method | ASQA | | Hotpot-QA | NQ | | Trivia-QA | | MuSiQue | ELI5 | |
|---|---|---|---|---|---|---|---|---|---|---|
| | Hit@1 | EM | EM | Hit@1 | EM | Hit@1 | EM | EM | ROUGE-L | BLEU |
| Llama-3.1-8B-Instruct-4K | | | | | | | | | | |
| BM25 | 45.00 | 19.84 | 36.25 | 40.75 | 30.66 | 84.75 | 26.17 | 5.75 | 15.90 | **6.56** |
| BGE | 68.50 | 31.47 | 43.25 | 59.00 | 44.59 | **92.25** | 27.50 | **10.00** | 15.87 | 6.30 |
| E5-Mistral | 62.50 | 28.51 | 38.50 | 56.50 | 41.73 | 90.00 | 27.05 | 9.00 | 15.77 | 5.85 |
| LongLLMLingua | 59.25 | 26.34 | 40.75 | 55.25 | 41.82 | 90.00 | 27.02 | 9.00 | **16.08** | 6.45 |
| JinaAI Reader | 53.50 | 23.14 | 34.00 | 47.25 | 34.41 | 84.75 | 24.83 | 6.75 | 15.80 | 5.65 |
| HtmlRAG | **71.75**† | **33.31**† | **43.75**† | **61.75**† | **45.90**† | 91.75† | **27.82**† | 8.75 | 15.51 | 5.84 |
| Llama-3.1-70B-Instruct-4K | | | | | | | | | | |
| BM25 | 49.50 | 21.95 | 38.25 | 47.00 | 35.56 | 88.00 | 25.63 | 9.50 | 16.15 | **6.99** |
| BGE | 68.00 | **30.57** | 41.75 | 59.50 | 45.05 | 93.00 | 27.04 | 12.50 | 16.20 | 6.64 |
| E5-Mistral | 63.00 | 28.75 | 36.75 | 59.50 | 44.07 | 90.75 | 26.27 | 11.00 | 16.17 | 6.72 |
| LongLLMLingua | 62.50 | 27.74 | 45.00 | 56.75 | 42.89 | 92.50 | **27.23** | 10.25 | 15.84 | 6.39 |
| JinaAI Reader | 55.25 | 23.73 | 34.25 | 48.25 | 35.40 | 90.00 | 25.35 | 9.25 | 16.06 | 6.41 |
| HtmlRAG | **68.50**† | 30.53† | **46.25**† | **60.50**† | **45.26**† | **93.50**† | 27.03 | **13.25**† | **16.33**† | 6.77† |

## 4.1 Datasets

We select six datasets, including: (1) ASQA [54]: a QA dataset consists of ambiguous questions that can be answered by multiple answers supported by different knowledge sources; (2) Hotpot-QA [67]: a QA dataset consists of multi-hop questions; (3) NQ [29]: A QA dataset containing real user's queries collected by Google; (4) Trivia-QA [24]: a QA dataset containing real user's questions; (5) MuSiQue [56]: A synthetic multi-hop QA dataset; (6) ELI5 [13]: A long-form QA dataset with questions collected from Reddit forum. We randomly sample 400 questions from the test set (if any) or validation set in the original datasets for our evaluation.

To simulate the real industrial web search environment, we require real web pages from the Web in HTML format as retrieved documents. However, the widely used Wikipedia search corpus mainly consists of pre-processed passages in plain text format. So, we apply Bing search API in the US-EN region to search for relevant web pages, and then we scrap static HTML documents through URLs in returned search results. We provide the URLs and corresponding HTML documents in our experiments for reproduction.

## 4.2 Evaluation Metrics

Our method aims to enhance the overall performance of RAG, so we evaluate the LLM's response as the end-to-end result. We choose different evaluation metrics for datasets according to their question-and-answer formats. For Hotpot-QA and MuSiQue, in which each question is annotated with a single short answer, we report Exact Match. For ASQA, NQ, and Trivia-QA, whose questions are annotated with several short answers, we report Exact Match and Hit@1. Hit@1 means at least one answer of the annotated answers finds the exact match in the LLM's response. ELI5 is annotated with long-form answers, and we report ROUGE-L [34] and BLEU [45].

## 4.3 Baselines

Since to the best of our knowledge, we are the first to take HTML as the format of retrieved knowledge in RAG systems, we compare HtmlRAG to baselines that conduct post-retrieval processes. These baselines are mainly based on plain text or Markdown format. We select three chunking-based refiners and uniformly follow the chunking method in LangChain framework [8]. The reranking compartment is plug-and-play and we use three different rerank models: (1) BM25 [51]: A widely used sparse rerank model; (2) BGE [65]: An embedding model, BGE-Large-EN with encoder-only structure; (3) E5-Mistral [60]: A embedding model based on an LLM, Mistral-7B [18], with decoder-only structure. Besides we select two abstractive refiners: (1) LongLLMLingua [19]: An abstractive model using Llama7B to select useful context; (2) JinaAI Reader [23]: An end-to-end light-weight LLM with 1.5B parameters fine-tuned on an HTML to Markdown converting task dataset.

## 4.4 Experimantal Settings

For a fair comparison, all end-to-end QA results are experimented with the latest open-source LLM, Llama-3.1-70B-Instruct and Llama-3.1-8B-Instruct [12] under a 4K context window. As for the implementation details of our method, we construct a block tree with a granularity of 256 words before pruning with the embedding model, and we construct a finer-grained block tree with a granularity of 128 words before pruning with the generative model. We choose BGE-Large-EN [65] as the embedding model for the HTML pruning. We choose a lightweight Phi-3.5-Mini-Instruct [1] with 3B parameters as the backbone for our generative model. The training data used in fine-tuning the generative model contains 2635 automatically constructed training samples ranging from 2K to 32K in length. More implementation details can be found in Appendix A.

Table 2: Results of HtmlRAG without pruning and baselines under the long-context setting. Hit@1 is the proportion of instances where at least one short answer matches. The best and second best results are in bold and underlined. The symbol † signifies that our method achieves superior results among baselines in a statistically significant manner (t-test, $p$-value < 0.05).

| Method | ASQA | | Hotpot-QA | NQ | | Trivia-QA | | MuSiQue | ELI5 | |
|---|---|---|---|---|---|---|---|---|---|---|
| | Hit@1 | EM | EM | Hit@1 | EM | Hit@1 | EM | EM | ROUGE-L | BLEU |
| Llama-3.1-8B-Instruct-128K | | | | | | | | | | |
| Vanilla HTML | 47.75 | 20.08 | 28.75 | 47.25 | 36.09 | 85.00 | 24.85 | 6.00 | **16.13** | 6.28 |
| Plain Text | 61.50 | **27.82** | 39.25 | **59.25** | **44.31** | **94.00** | **28.23** | 7.75 | 16.02 | **6.35** |
| Markdown | **61.75** | 26.70 | 37.50 | 57.50 | 42.85 | 91.50 | 26.67 | 7.50 | 16.12 | 5.91 |
| HtmlRAG w/o Prune | 61.00 | 26.70$^{†}$ | **39.50**$^{†}$ | 59.00$^{†}$ | 43.46$^{†}$ | 92.00$^{†}$ | 27.50$^{†}$ | **8.75**$^{†}$ | 15.62 | 5.87 |
| Llama-3.1-70B-Instruct-128K | | | | | | | | | | |
| Vanilla HTML | 44.00 | 17.52 | 28.00 | 46.75 | 36.06 | 81.50 | 22.58 | 3.25 | 15.69 | 5.16 |
| Plain Text | **59.75** | 25.16 | 41.00 | **59.75** | **44.11** | 93.50 | 26.75 | 8.75 | 16.88 | **7.44** |
| Markdown | 56.00 | 24.00 | 39.00 | 57.00 | 42.00 | 92.00 | 26.43 | 8.25 | **16.91** | 6.74 |
| HtmlRAG w/o Prune | 58.75$^{†}$ | **25.28**$^{†}$ | **42.25**$^{†}$ | 58.00$^{†}$ | 43.65$^{†}$ | **95.00**$^{†}$ | **27.21**$^{†}$ | **10.75**$^{†}$ | 16.57 | 6.32 |

## 4.5 Experimental Results

Main experimental results are demonstrated in Table 1. Our method, HtmlRAG meets or exceeds the baselines across all metrics on the six datasets. This demonstrates the effectiveness of HTML pruning. Additionally, we make the following observations:

(1) For chunking-based refiners, we followed LangChain's [8] chunking rule, which chunks according to HTML tag headings (h1, h2, etc.). Although this chunking strategy considers certain HTML structures, it does not utilize the structural information as effectively as our method. Moreover, converting the final output to plain text still results in a loss of HTML structural and semantic information. Among the three rerankers we applied, the sparse retriever BM25 is inferior to two dense retrievers. Among two dense retrievers, the encoder-based BGE performs better than the decoder-based e5-mistral, despite the latter having more parameters.

(2) Among the abstractive refiners, LongLLMLingua is not optimized for HTML documents, so its extraction ability is affected when dealing with HTML. Additionally, the plain text output loses structural information, resulting in inferior performance compared to our method. The JinaAI-reader generates the refined Markdown given the HTML input. However, token-by-token decoding with long input and output lengths is not only challenging for end-to-end generative models, but also has high computational cost.

## 4.6 Further Analysis

*4.6.1 The Effectiveness of HTML Cleaning.* To validate the priority of HTML as the format of retrieved knowledge, we compare our HTML cleaning module, namely the results of HtmlRAG without pruning, with other rule-based cleaning strategies, including (1) Vanilla HTML; (2) Plain Text: The plain text extracted with an on-the-self package BeautifulSoup [50]; (3) Markdown: The Markdown converted by an on-the-self converter Markdownify [2]. Additional experiments on token count show that HTML-Clean drops over 94.07% tokens of the original HTML, while the number for plain text and Markdown conversion are 96.71% and 90.32% respectively.

The cleaned HTML is still long, so we conduct experiments under a long-context setting (128K), as shown in Table 2. When HTML is taken as the format of external knowledge, HtmlRAG without pruning meets or outperforms plain text and Markdown on most datasets, demonstrating its validity. Besides, we make the following observations: (1) Unprocessed HTML documents contain a large amount of irrelevant content, so all cleaning algorithms show improvements over vanilla HTML. (2) A more capable LLM (70B) performs better than a less capable one (8B) when taking HTML as the format of external knowledge. This indicates that more powerful models are better at understanding the complex information within HTML.

*4.6.2 Ablation Study.* We conduct ablation studies to demonstrate the effectiveness of each component in HtmlRAG, including block tree construction (Block Tree), HTML pruning with the embedding model (Prune-Embed), and HTML pruning with the generative model (Prune-Gen). From the results in in Table 3, we can see: (1) In the ablation study for block tree construction, we use the DOM tree instead of the block tree. Units in the DOM tree are so fragmented that the embedding model fails to capture sufficient semantic features, thus causing a drop in performance. The performance of the generative model is also affected due to the increase in the length of block paths. (2) In the ablation study for pruning with the embedding model, we only use the generative model to prune the cleaned HTML. Without the basically pruned HTML by the embedding model, the input to the generative model becomes very long (exceeds 32K), resulting in high computational costs and poor performance. (3) In the ablation study for pruning with the generative model, we only use the embedding model to prune the cleaned HTML. The result is inferior compared to the further pruned HTML using the generative model, because the embedding model's global understanding and ability to process finely-grained block trees are inferior to the generative model.

*4.6.3 Impact of Block Tree Granularity.* The most critical hyperparameter in HTML pruning is granularity. A coarse granularity reduces the flexibility of pruning, while a fine granularity makes

Table 3: Ablation studies for HtmlRAG.

| Method | ASQA | | Hotpot-QA | NQ | | Trivia-QA | | MuSiQue |
|---|---|---|---|---|---|---|---|---|
| | Hit@1 | EM | EM | Hit@1 | EM | Hit@1 | EM | EM |
| HtmlRAG | **68.50** | **30.53** | **46.25** | **60.50** | **45.26** | **93.50** | **27.03** | **13.25** |
| *w/o* Block Tree | 59.50 (9.00%↓) | 25.50 (5.03%↓) | 40.25 (6.00%↓) | 56.25 (4.25%↓) | 42.07 (3.19%↓) | 92.00 (1.50%↓) | 26.59 (0.44%↓) | 8.00 (5.25%↓) |
| *w/o* Prune-Embed | 56.75 (11.75%↓) | 24.05 (6.48%↓) | 37.50 (8.75%↓) | 49.50 (11.00%↓) | 37.27 (7.99%↓) | 91.75 (1.75%↓) | 26.02 (1.01%↓) | 9.75 (3.50%↓) |
| *w/o* Prune-Gen | 62.00 (6.50%↓) | 26.74 (3.79%↓) | 38.75 (7.50%↓) | 57.75 (2.75%↓) | 42.91 (2.35%↓) | 89.50 (4.00%↓) | 25.55 (1.48%↓) | 7.00 (6.25%↓) |

Figure 5: Experimental results for the impact of block tree granularity. The results of Prune-Embed and Prune-Gen are represented in a bar chart, with a red dashed horizontal line indicating the performance of the strong baseline method, chunking-based refiner with BGE (BGE-Chunk-Rerank).

it difficult to extract text embeddings for small blocks, and leads to overly long block paths for the generative model, so we need to find balancing points. In Figure 5, we experiment with HTML pruning under different granularity ranging from 64 to 512 words, and compare their result with a strong baseline. Prune-Embed stands for using the basically pruned HTML by the embedding model, and Prune-Gen stands for using the finely pruned HTML by the generative model. It can be observed that the generative model adapts to a finer granularity than the embedding model and generally outperforms the embedding model. This validates the rationality of our two-stage pruning method.

*4.6.4 Light Weight HTML Pruning.* To show that our HTML pruning method does not significantly increase the computational cost despite using an LLM with 3B parameters, we conduct an efficiency analysis. Table 4 shows the computational cost of our method compared to the baseline and the cost of the LLM's inference. We can see that the HTML pruning with the embedding model still maintains a similar computational cost to the chunking-based refiner. The computational cost of the generative model is a bit higher than the baseline but still much lower than the cost of the LLM for chatting. Additional experiments show that there are over 45% of nodes that can be skipped, explaining the little increase in the generative model's computational cost.

Analysis of token counts shows the average token count for all retrieved knowledge in HTML format is 1.6M, suppose we retrieve 20 HTML documents. HTML cleaning reduces the token count to 135K, HTML pruning based on text embedding reduces it to 8K, and generative HTML pruning reduces it to 4K. In typical RAG scenarios, since the computational cost of HTML pruning is much less than the inference cost of the LLM, we recommend using complete HTML pruning to achieve the best results. Meanwhile, in

| Result Length | # Params | Storage | # In-Tokens | # Out-Tokens |
|---|---|---|---|---|
| BGE | 200M | 2.5G | 93.54K | 740.3 |
| Prune-Embed | 200M | 2.5G | 152.5K | 2653 |
| Prune-Gen | 3B | 7.2G | 6750 | 28.70 |
| LLM Chat | 70B | 131G | 3661 | 182.9 |

Table 4: Analysis of inference cost on ELI5 dataset We compare the chunking-based refiner using BGE (BGE), the two HTML pruning steps basing on the text embedding (Prune-Embed) and the generative model (Prune-Gen) in HtmlRAG, and LLM chatting (LLM Chat) by model parameters, storage, average input tokens, and average output tokens.

some resource-limited scenarios where the cost of HTML pruning is also a concern, we suggest using only the basically pruned HTML from the embedding model. Basically pruned HTML also yields performance that meets or surpasses the chunking-based refiner, as we can observe from Figure 5.

## 5 Conclusion and Future Work

In this work, we propose taking HTML as the format of external knowledge in RAG systems. To tackle the additional tokens brought by HTML, we design HTML cleaning and HTML pruning to shorten HTML while retaining key information. Experiments show that HtmlRAG outperforms existing post-retrieval processes based on plain text, and validates the priority of HTML as the format of retrieved knowledge. Moreover, this work opens up a new research direction and provides a simple and effective solution. We believe as LLMs become more powerful, HTML will be more suitable as the format of external knowledge. We also hope that future works will propose better solutions for processing HTML in RAG systems.

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

## A  Generative Model Training Details

Here we introduce several critical hyper-parameters that define the training process of the generative model. The model's max training context window is set to 35000 tokens. The model is trained for 3 epochs. The training is conducted on 4 computing nodes, with 32 Nvidia A800 GPUs, each having 80G memory. To manage memory usage and computational efficiency, *per_device_train_batch_size* is set to 1, while *gradient_accumulation_steps* is set to 8, effectively simulating a larger batch size during backpropagation.

For parallelism, *seq_parallel_size* is set to 8, indicating that the model will distribute its computations across 8 devices if available. The *learning_rate* is set to 2e-5, striking a balance between rapid convergence and avoiding divergence. The learning rate scheduler (*lr_scheduler_type*) is set to 'constant', meaning the learning rate remains unchanged throughout the training unless manually adjusted. For optimization, the Adam optimizer parameters (*adam_beta*1, *adam_beta*2, and *adam_epsilon*) are chosen as 0.9, 0.98, and 1e-8 respectively, to ensure stable gradient updates. The *max_grad_norm* is set to 1.0 to prevent exploding gradients by clipping them if they exceed this norm. A weight decay (*weight_decay*) of 1e-4 is used to regularize the model and prevent overfitting. A *warmup_ratio* of 0.01 indicates that the learning rate will be gradually increased during the initial 1% of the training process before settling at the base learning rate. *gradient_checkpointing* is enabled to save memory at the cost of increased computation time.

DeepSpeed is configured for efficient distributed training. For ZeRO optimization (*zero_optimization*), stage 3 is selected, which represents the highest level of parameter partitioning and offloading. Gradient clipping (*gradient_clipping*) is set to 1.0, ensuring that the gradients do not grow too large, thus preventing potential issues like exploding gradients. The *wall_clock_breakdown* option is set to false, indicating that DeepSpeed will not provide a detailed breakdown of the training time spent on different components of the training loop, which can be useful for profiling but may add some overhead. Mixed precision training using bfloat16 is set to "auto", indicating that DeepSpeed will decide whether to use bfloat16 based on the capabilities of the system and the requirements of the model.

## B  Key Algorithms

In this appendix section, we present all the algorithms mentioned in the main text using pseudo code, including the algorithm for constructing the block tree, the pruning algorithm using the embedding model, and the pruning algorithm using the generative model.

To make it clear, we first define elements under a certain node as follows: All sorts of elements under the node are referred to as *node.content*; Text wrapped by child tags is referred to as *node.children*; Text directly attached to the node is referred to as *node.text*. We show an example accordingly in Figure 6. To discriminate between children with the same HTML tag, we append a number to the end of the original tag name. For example, two children with the same "<div>" tag are renamed as "<div1>" and "<div2>".

The block tree construction algorithm is demonstrated in Algorithm 1, which transforms a DOM Tree $T$ into a Block Tree $T'$. In the block tree, a block is the smallest unit that be pruned in

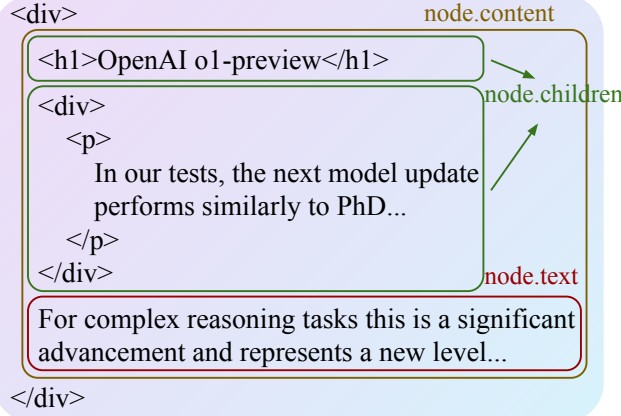

**Figure 6: Node content explained**

subsequent steps. We use a breadth-first algorithm to traverse all nodes in the DOM tree. Leaf nodes that are visited are directly considered as blocks. If the total number of tokens of all content under a node is less than the number we set (*maxWordss*), we merge all the content of the node and consider it as a block. Otherwise, we check the content of the node. The node's children are to be visited in subsequent steps. The node's text will be considered as a block. It is noteworthy that if there are only children but no text under the node, it will not be considered as a block. This algorithm merges fragmented nodes as a block, until the number of tokens exceeds *maxWords*.

Another key algorithm is greedy block pruning, as demonstrated in Algorithm 2. We greedily delete the block with the lowest score until the length of the HTML document meets the context window we set. To elaborate, when deleting a block, if the block is a leaf node, we delete the block directly. Otherwise, if the block consists of directly attached text under a parent node, we delete only those text. After a block is deleted, the algorithm recursively checks if the parent node is empty. If the parent node is empty, it is to be deleted.

The last key algorithm is token probability calculation, as demonstrated in Algorithm 3. We use a depth-firth algorithm to traverse tokens in the token tree so that tokens visited sequentially share the longest prefix sequences. The probability of the root token and singleton child tokens are directly set to 1.0, and does not require calculation.

Received 20 February 2007; revised 12 March 2009; accepted 5 June 2009

**Algorithm 3** Token Probability Calculation

1: **procedure** TRAVERSETOKENTREE
2:     Declare a queue $nodeStack$
3:     $t_0 \leftarrow$ root node of $T$
4:     $t_0.prob \leftarrow 1.0$              ▷ Set probability of $R$ as 1.0
5:     Push $t_0$ into $nodeStack$
6:     **while** $nodeStack$ is not empty **do**
7:         $t_{n-1} \leftarrow$ Pop from $nodeQueue$
8:         $children \leftarrow$ Expand children of node $p : (t_n^0, t_n^1, \cdots)$
9:         **if** $|children| = 0$ ($p$ is a leaf node) **then**
10:             **continue**
11:         **else if** $|children| = 1$ **then**
12:             $t_n^0.prob \leftarrow 1.0$
13:             Push the singleton child $t_n^0$ into $nodeQueue$
14:         **else**
15:             $prefix \leftarrow \{input, t_0, \ldots, t_{n-1}\}$
16:             **for** each $t_n^i$ in $children$ **do**
17:                 $t_n^i.prob \leftarrow \dfrac{exp(Logits(t_n^k))}{\sum_{i=0}^{K} exp(Logits(t_n^i))}$
18:                 Push $t_n^i$ into $nodeQueue$
19:             **end for**
20:         **end if**
21:     **end while**
22: **end procedure**

**Algorithm 1** Construct Block Tree $T'$ from DOM Tree $T$

1: **procedure** CONSTRUCTBLOCKTREE($T$)
2:     Declare a queue $nodeQueue$
3:     $R \leftarrow$ root node of $T$
4:     Enqueue $R$ into $nodeQueue$
5:     **while** $nodeQueue$ is not empty **do**
6:         $node \leftarrow$ Dequeue from $nodeQueue$
7:         **if** $node$ is a leaf node **then**
8:             $node.block \leftarrow$ node.content
9:             $node.isLeaf \leftarrow$ True
10:         **else**
11:             **if** $node.content < maxTokens$ **then**
12:                 Merge descendant nodes of $node$
13:                 $node.block \leftarrow$ node.content
14:                 $node.isLeaf \leftarrow$ True
15:             **else**
16:                 Expand children of $node$
17:                 **for** each child of $node$ **do**
18:                     Enqueue child into $nodeQueue$
19:                 **end for**
20:                 **if** $node.text$ is not empty **then**
21:                     $node.block \leftarrow$ node.text
22:                     $node.isLeaf \leftarrow$ False
23:                 **end if**
24:             **end if**
25:         **end if**
26:     **end while**
27:     **return** $T$
28: **end procedure**

**Algorithm 2** Greedy Block Pruning

1: **procedure** GREEDYBLOCKTREEPRUNING($\mathbf{T}$)
2:     $nodes \leftarrow$ all nodes with blocks from $\mathbf{T}$
3:     **for** each $node$ in $nodes$ **do**
4:         $node.score \leftarrow Rel(q, node.block)$   ▷ calculate semantic similarity between node $node$ and user request
5:     **end for**
6:     Sort $nodes$ by key $node.score$ in ascending order
7:     **for** each $node$ in $nodes$ **do**
8:         **if** $node.isLeaf$ **then**
9:             $parent \leftarrow node.parent$
10:             delete $node$
11:             **while** $parent.content$ is empty **do**
12:                 $parent \leftarrow parent.parent$
13:                 delete $parent$
14:             **end while**
15:         **else**
16:             delete $node.text$
17:         **end if**
18:     **end for**
19: **end procedure**

