# OpenReview forum: "HtmlRAG: HTML is Better Than Plain Text for Modeling Retrieved Knowledge in RAG Systems"
_ACM.org/TheWebConf/2025/Conference — WWW 2025 Oral_

### Official Review · Reviewer_n7h6 · 2024-11-25

**Novelty:** 4
**Technical Quality:** 5

**Review:**

The manuscript proposes HtmlRAG, an improved Retrieval-Augmented Generation (RAG) system that leverages HTML format instead of plain text to retain the structural and semantic information of documents. Through HTML cleaning, compression, and DOM-based pruning strategies, HtmlRAG effectively addresses the issue of overly long HTML documents, outperforming traditional methods across six question-answering datasets. The manuscript presents innovative ideas, a well-organized approach, clear articulation, and a rigorous experimental design.

The advantages of this approach include:
1. Using HTML as the input format retains inherent structural and semantic information, such as headers and table structures.
2. The proposed HTML pruning algorithm balances efficiency and effectiveness by constructing an adjustable granularity block tree instead of using the DOM tree, reducing computational costs while ensuring processing efficiency through block pruning based on text embeddings and generative fine-grained block pruning.
3. Extensive baseline comparisons on multiple datasets demonstrate significant improvements, supported by theoretical analysis in ablation studies.

My main concern is about the results shown in table 2. Even with a more capable LLM (Llama 70B), the proposed HtmlRAG outperforms other methods on only 5/10 experiments. Using plain text yields the best results on 4/10 experiments. It would be helpful if the authors could provide more elaboration on why it is worthwhile of doing the html cleaning, block-based pruning, instead of doing more tricks on plaintext.

**Questions:**

Q1: given the results in table 2, how to convince others that the benefit of HtmlRAG is not marginal.
Q2: Some technical details on implementation or training methods of the HTML cleaning algorithm would be helpful.

**Reviewer Confidence:**

3: The reviewer is confident but not certain that the evaluation is correct

**Scope:**

4: The work is relevant to the Web and to the track, and is of broad interest to the community

---

### Official Review · Reviewer_zv2a · 2024-11-25

**Novelty:** 4
**Technical Quality:** 5

**Review:**

Summary

This paper proposes a method to enhance the existing Retrieval-Augmented Generation (RAG) pipeline by using HTML as the document format. The authors argue that using plain text leads to the loss of structural and semantic information present in web pages. To address the challenge of content length associated with HTML documents, the paper introduces two modules: HTML Cleaning and HTML Pruning, which streamline and maintain the DOM tree while minimizing semantic loss. In the HTML Pruning module, the authors train a lightweight large language model (LLM) to generate HTML tags and assign scores to each block in the DOM tree, facilitating block pruning. Experiments on six question-answering datasets demonstrate that the proposed method strikes a balance between efficiency and effectiveness.

Strengths
1. The HTML Cleaning and HTML Pruning strategies proposed in this paper are innovative to some extent, particularly the design of using LLMs to refine the DOM tree.
2. The figure illustrations in the paper are clear and intuitive, effectively supporting the description of the methodology. Additionally, the algorithmic workflows supplement unclear areas, making the approach easier to understand.
3. The paper provides a thorough review of related work and is written in a clear manner.

Weaknesses
1. Unclear Baseline Comparisons. The implementation and comparison of baseline algorithms are not well-explained. For instance, it is unclear whether the main results in Table 1 use HTML or plain text as the source of external knowledge. If HTML is used without preprocessing, the comparison may be unfair, as raw HTML tags and CSS styles could occupy a significant portion of tokens.
2. Incomplete Efficiency Experiments. Table 4 in the paper only reports token and storage consumption but does not summarize inference latency. While converting HTML pages to plain text is highly efficient, the proposed Path-aware Generative Model relies on LLMs to process HTML, potentially introducing efficiency risks. Thus, it is necessary to compare inference latency to substantiate the claim that the method “strikes a balance between efficiency and effectiveness.”
3. Limited Generalizability of the Proposed Pruning Method. The Generative Fine-Grained Block Pruning method is query-specific, which limits its generalizability. As shown in Figure 4, the LLM-based Block Pruning prunes HTML based on a specific input question, making it unsuitable for handling HTML pages for arbitrary queries. Additionally, this approach may further impact efficiency since pruning operations require LLM inference for every query and HTML page.

**Questions:**

1. In line 457, the paper mentions “calculate the exact match score for the content within blocks with the gold answer.” How is this achieved?
2. The authors claim that current LLMs struggle to handle lengthy HTML inputs but use Phi-3.5-Mini-Instruct, which supports 128K context, for pruning. Does this contradict the claim?
3. In Table 2, performance on certain datasets (e.g., ASAQ and NQ) decreases when switching from Llama-3.1-8B to Llama-3.1-70B. What might be the reasons for this phenomenon?
4. Section 4.2 mentions using web pages crawled from Bing for supplementation but does not elaborate on the specifics. For instance, how many pages were used as knowledge sources per question?
5. Typo. Line 682: “Experimantal” should be corrected to “Experimental.”

**Reviewer Confidence:**

3: The reviewer is confident but not certain that the evaluation is correct

**Scope:**

3: The work is somewhat relevant to the Web and to the track, and is of narrow interest to a sub-community

---

### Official Review · Reviewer_n5QX · 2024-11-29

**Novelty:** 6
**Technical Quality:** 6

**Review:**

Authors propose taking HTML, instead of just plain text, as the format of external information to be used by RAG systems, since this allows to retain structural and semantic information in web pages which would otherwise be lost. They undertake the challenging task of properly reducing the large size of HTML documents before feeding them into LLM for results augmentation, employing rather sophisticated cleaning and pruning techniques. The proposed methodology appears to be sound, and it is supported by an extensive performance evaluation section, including six datasets of questions and answers, comparison with other approaches, and an ablation study. Overall, I find the work original and of good quality, and definitely significant for the the Web Conf and the chosen  track. Some minor suggestions to improve the text are listed in the questions section below.

**Questions:**

I could not quite understand how the methodology works in the case of a collection of documents with heterogenous sizes. Are all documents merged together into a single one before processing? Then isn't this somehow "unfair" accross the original sources? (a small one might be as valuable as a large one, but it undergoes the same (unique) reduction process resulting from the overall ensemble). Please clarify.

I also disagree with the comment on the results of Table II, where it is said that (lines from 782) the larger (70B) model performs
better than the 8B one: in 4 cases out of 10 results get worse, despite the much larger model complexity...! please clarify

Some statements are unclear or should be fixed/improved:
101: to the loss of -> to a significant loss of
129: "The noisy tokens." (orphan)
322: "and it's needless to involve semantic features": -->unclear
329: "and are not willing to lose any information...": -> to be fixed
355: numerous nood... -> the large number of nodes
577 (also 616): we simulate the real industrial working scenario? -> we simulate a realistic scenario

**Reviewer Confidence:**

3: The reviewer is confident but not certain that the evaluation is correct

**Scope:**

4: The work is relevant to the Web and to the track, and is of broad interest to the community

---

### Official Review · Reviewer_sVZq · 2024-12-02

**Novelty:** 5
**Technical Quality:** 5

**Review:**

This paper presents HtmlRAG, a RAG method that utilized HTML instead of plain text as the format of retrieved knowledge in RAG. The paper addresses the challenge of using HTML. To solve this challenge, the authors propose a pruning process on HTML, and finally striked a balance between efficiency and effectiveness. The paper includes experimental results that demonstrate the superiority of using HTML in RAG systems.

Strengths

1. The paper is well-written and clearly presented; The paper introduces a novel method that use html as a format and also introduces how to use HTML efficiently to mitigate the effects of noise

2. An interesting research question: the idea of using HTML instead of plain text for RAG, preserving valuable structural and semantic information typically lost during conversion. In general, LLM has a stronger understanding and generation ability for text than other data formats due to the corpus in the pre-training phase. This paper proposes new views on RAG issues.

3. Practicality: The proposed pruning strategies, including HTML cleaning and block-tree-based pruning, are well-motivated and applicable to a variety of RAG systems.

Weaknesses

1. Limited Error Analysis. The paper provides minimal analysis of failure cases. For example, when the model fails to capture relevant information from pruned blocks, it remains unclear whether the issue lies with the pruning process, the embedding model, or the generative model.

2. Limited Discussion on Plain Text Limitations. While the paper criticizes the use of plain text for losing structural and semantic information, it does not provide a detailed quantitative analysis comparing plain text performance to HTML under controlled settings. This weakens the argument for why HTML is inherently better, which is a question worth exploring.

3. Insufficient Explanation of Pruning Metrics. The generative pruning method depends heavily on calculated relevance scores for HTML blocks, but the scoring mechanism, especially for nuanced content, is not detailed.

**Questions:**

1. Are there specific domains or datasets where plain text might still outperform HTML due to simpler content or fewer structural features?

2. How does the block-tree construction handle interdependent blocks (e.g., references or links to other parts of the document)? Does pruning ever break these dependencies?

3. Have you tested HtmlRAG on non-QA tasks, such as summarization or classification? If so, how does its performance compare to other state-of-the-art approaches?

4. How does the size of the embedding model and generative model impact HtmlRAG's effectiveness? Are there specific thresholds where diminishing returns are observed?

**Reviewer Confidence:**

3: The reviewer is confident but not certain that the evaluation is correct

**Scope:**

4: The work is relevant to the Web and to the track, and is of broad interest to the community

---

### Official Review · Reviewer_CW8s · 2024-12-02

**Novelty:** 4
**Technical Quality:** 4

**Review:**

Summary:
This paper presents HtmlRAG, a method for improving RAG systems by using HTML rather than plain text for retrieved knowledge. It addresses the challenge that HTML’s rich structural information, such as headings and tables, is often lost when converted to plain text. HtmlRAG incorporates an HTML cleaning process to remove unnecessary elements and a two-step pruning method: first, irrelevant HTML blocks are discarded based on their relevance to the query, then a generative model refines the selection of content. The method is evaluated on six diverse QA datasets and compared with baselines that use plain text or Markdown. The results show that using HTML can enhance RAG systems, provided that effective pruning is employed to handle the additional complexity of HTML documents.
Strengths:
1.	The paper provides a clear explanation of the challenges in using HTML in RAG systems and the solutions implemented in HtmlRAG.
2.	The paper's exploration of HTML as a knowledge format in RAG systems addresses a gap in current research, offering potential for improving the quality of knowledge retrieval.
3.	The method is presented logically and well-written.
Weaknesses:
1.	The cleaning and pruning steps, which are central to reducing the size of the HTML document for efficient processing, are not thoroughly explained. The paper briefly mentions cleaning and block pruning, but the specific criteria or techniques used to identify and remove irrelevant HTML blocks are not detailed. Additionally, there is no formal evaluation or benchmarking of these steps to measure their impact on the retention of meaningful content. The authors are recommended to include a formal analysis or case study that compares the original HTML document, the pruned HTML, and the final output to demonstrate the effectiveness and potential risks of pruning.
2.	If the generative model is not well-calibrated or tested across different types of documents, it may fail to consistently identify the most relevant content, reducing the effectiveness of the method.
3.	The trade-off between reducing document size and maintaining information richness may lead to scenarios where essential context is lost, impacting the quality of the answer generation.

**Questions:**

1.	How exactly do the authors determine which HTML blocks are "useless" or irrelevant during the cleaning process? What criteria are used to select the blocks to be pruned? Are these criteria consistent across different types of HTML documents?
2.	How do the authors address the potential for the generative model to misinterpret or overlook important HTML blocks during the pruning process?
3.	How do the authors measure or quantify the loss of information during the pruning process?  Is there a risk that the pruning strategy could discard content that is indirectly relevant to the query but crucial for accurate answer generation?

**Reviewer Confidence:**

3: The reviewer is confident but not certain that the evaluation is correct

**Scope:**

3: The work is somewhat relevant to the Web and to the track, and is of narrow interest to a sub-community